# Study on the Effectiveness of a Copper Electrostatic Filtration System “Aerok 1.0” for Air Disinfection

**DOI:** 10.3390/ijerph21091200

**Published:** 2024-09-10

**Authors:** Roberto Albertini, Maria Eugenia Colucci, Isabella Viani, Emanuela Capobianco, Michele Serpentino, Alessia Coluccia, Mostafa Mohieldin Mahgoub Ibrahim, Roberta Zoni, Paola Affanni, Licia Veronesi, Cesira Pasquarella

**Affiliations:** 1Department of Medicine and Surgery, University of Parma, Via Gramsci 14, 43126 Parma, Italy; roberto.albertini@unipr.it (R.A.); mariaeugenia.colucci@unipr.it (M.E.C.); isabella.viani@unipr.it (I.V.); michele.serpentino@studenti.unipr.it (M.S.); roberta.zoni@unipr.it (R.Z.); paola.affanni@unipr.it (P.A.); licia.veronesi@unipr.it (L.V.); 2Geriatric-Rehabilitation Department, University Hospital-Azienda Ospedaliero-Universitaria di Parma, Via Gramsci 14, 43126 Parma, Italy

**Keywords:** air contamination, microorganisms, pollen, particles, sampling, air disinfection

## Abstract

Background: Bioaerosols can represent a danger to health. During SARS-CoV-2 pandemic, portable devices were used in different environments and considered a valuable prevention tool. This study has evaluated the effectiveness of the air treatment device “AEROK 1.0^®^” in reducing microbial, particulate, and pollen airborne contamination indoors, during normal activity. Methods: In an administrative room, airborne microbial contamination was measured using active (DUOSAS 360 and MD8) and passive sampling; a particle counter was used to evaluate particle concentrations; a Hirst-type pollen trap was used to assess airborne pollen and *Alternaria* spores. Statistical analysis was performed using SPSS 26.0; *p* values < 0.05 were considered statistically significant. Results: The airborne bacterial contamination assessed by the two different samplers decreased by 56% and 69%, respectively. The airborne bacterial contamination assessed by passive sampling decreased by 44%. For fungi, the reduction was 39% by active sampling. Airborne particles (diameters ≥ 1.0, 2.0 μm) and the ratio of indoor/outdoor concentrations of total pollen and *Alternaria* spp. spores significantly decreased. Conclusions: The results highlight the effectiveness of AEROK 1.0^®^ in reducing airborne contamination. The approach carried out represents a contribution to the definition of a standardized model for evaluating the effectiveness of devices to be used for air disinfection.

## 1. Introduction

A part of aerosol consists of particles of biological origin (bioaerosol). Airborne particulate is a topic of great interest in many areas of human activity, as it represents a danger to human health and the integrity of materials [1,2,3,4,5,6,7,8,9,10,11]. In indoor environments, the concentrations of particulate may be higher than outdoor depending on the activity taking place there, the number of occupants, construction characteristics, materials and furnishings characteristics, and the possible presence of air treatment systems. Indoor air quality plays an important role in human health if we consider that we spend up to 90% of our time indoors (domestic, work, leisure, transport, etc.) [12,13,14].

World Health Organization has described a combination of symptoms associated with staying in a building, without specific causes, named sick building syndrome (SBS) [15]. It is a disease with multifactorial etiology, with non-specific symptoms, such as headache, drowsiness, difficulty concentrating, memory issues, asthenia, nausea, eye irritation, nose, throat, asthmatic respiratory problems, skin rashes, dryness and irritation of the throat and gastrointestinal manifestations. The presence of particulate matter, especially of fungal origin, together with unfavorable microclimatic parameters is often considered one of the causes of the symptoms [16,17,18,19]. Indoor air quality also can affect mental well-being [20].

To achieve high-quality indoor air, various air treatment systems, with different mechanisms of action, have been used, including HVAC (heating, ventilation, and air conditioning) systems equipped with HEPA (high-efficiency particulate air) filters, UV-C treatment, electrostatic filtration, chemical treatment, ozone treatment, plasma treatment, or a combination of some of the different methods [21,22,23,24,25,26,27,28,29,30,31,32,33,34,35,36,37,38,39,40,41,42,43,44,45,46,47].

Ventilation and air disinfection were a valuable prevention tool during the SARS-CoV-2 pandemic, and portable air purification devices have also been used [48,49,50,51,52] in different kinds of environments. In this regard, it is of particular importance that air purifying devices are also assessed based on their effectiveness against different kinds of bioaerosols [53] and could be useful to emphasize that the use of air purifiers does not replace natural ventilation and air exchange outdoors/indoors.

This study aimed to evaluate the effectiveness of the air treatment device “AEROK 1.0^®^” (AERSAFE s.r.l., Trento, Italy), hereinafter AEROK, in reducing microbial, particle, and pollen airborne contamination in working environments.

## 2. Methods

### 2.1. Features of the Device AEROK

AEROK treats 1250 m^3^/h of air. Air disinfection is carried out by a copper electro-filter. The filter undergoes periodic self-washing with distilled water, which removes solid particles present on the accumulation plates, restoring the initial filtration capacity. The water is constantly disinfected by UV radiation. The noise level is lower than 40 dB.

### 2.2. Study Site

AEROK was installed in a room used for administrative purposes at the Laboratory of Hygiene and Public Health, of the Department of Medicine and Surgery of the University of Parma. The room had the following dimensions: 4.50 m height, 5.25 m length, 3.50 m width of 18.25 m^2^ surface area, and 82.68 m^3^ volume. The room was occupied by 2–3 people.

### 2.3. Microbial Sampling

The airborne concentration of microorganisms (cfu/m^3^, colony forming units per cubic meter of air) was measured by active sampling, and the sedimentation rate of microorganisms on surfaces (cfu/plate/time) was measured by passive sampling [54,55,56,57,58,59,60,61].

#### 2.3.1. Active Sampling

Two PBI (VWR)—DUOSAS 360 samplers with 55 mm diameter RODAC plates (Replicate Organism Detection and Agar Counting) for bacterial and fungal evaluation were used. A Sartorius MD8 Airport Portable Air Sampler, with gelatine filter membranes, for bacterial evaluation was used; after sampling the gelatine membranes were put on Petri dishes of 9 cm diameter. The results were expressed as cfu/m^3^.

#### 2.3.2. Passive Sampling

Petri dishes of 9 cm in diameter were exposed at a height of 1 m for 1 h to determine the Index of Microbial Air (IMA) for bacterial and fungal evaluation.

#### 2.3.3. Cultural Conditions

TSA medium (tryptic soy agar) and SDA medium (Sabouraud dextrose agar) were used, to determine bacterial and fungal contamination, respectively. TSA plates were incubated for 48 h at 36 ± 1 °C and SDA plates were incubated for 120 h at 25 ± 1 °C.

#### 2.3.4. Microbial Monitoring Plan

Microbial monitoring was carried out for five working days, with samplers and plates positioned according to the scheme shown in Figure 1 and Figure 2. The study involved two hours of sampling with AEROK turned off; then AEROK ran for one hour, without sampling. The sampling started again at the 2nd hour of the AEROK running over the next two hours. The sampling took place during regular working activity, and the number of door openings in the room was recorded during the sampling time to assess the working traffic.

For one-hour sampling, serial active microbiological sampling with the suction of 125 L of air four times was performed (every 20 min, at 0, 20, 40, and 60 min), and settle plates were exposed in the studied room. Furthermore, an additional passive sampling point was placed in the corridor adjacent to the room, where air treatment was absent.

### 2.4. Particle Sampling

The particle concentration measurements were carried out in the center of the room by using a particle counter LASAIR III 5100 (Particle Measuring System) according to ISO 14644-1 and 14644-2 [61,62] (Figure 1).

#### Particles Sampling Plan

Two hundred measures each were taken (one per minute) by continuously sampling 100 L of air, with AEROK off and then with AEROK on, in the absence of people.

The particle counter LASAIR III 5100, with a flow rate of 100 L/minute and channels for sampling particles ≥0.5; 1.0; 2.0; 5.0; 10.0; 25.0 μm, was certified and validated according to ISO 21501-4:2018 [63].

### 2.5. Aerobiological Sampling of Pollen and Fungal Spores

The sampling of airborne pollen and fungal spores was carried out using a volumetric Hirst-type pollen trap (7-day Burkard spore trap) [64,65]. The sampler worked continuously for two weeks with the change in the sampling tape on the seventh day. After sampling, the tape, previously treated with silicone, was mounted with gelatine on slides, one for each day of sampling, stained with fuchsin, and observed under a microscope for the recognition and counting of total pollen and fungal spores of *Alternaria* spp. The values obtained were expressed as spores per m^3^.

The concentration of indoor (I) total pollen and fungal spores of *Alternaria* spp. were compared to those outdoor (O) (kindly provided by the Regional Agency for Prevention, Environment and Energy of Emilia-Romagna, ARPAE). The indoor-to-outdoor concentration ratio (I/O) result was used as a reference value to compare the results obtained with AEROK off and with AEROK on.

#### 2.5.1. Sampling Points

Inside the room, the sampling of total pollen and fungal spores of *Alternaria* spp. was performed according to the scheme shown in Figure 1.

#### 2.5.2. Pollen and Fungal Spores Sampling Plan

The sampling of total pollen and *Alternaria* spores was carried out simultaneously, keeping the windows constantly open to allow the aerospora entry from outside, for 7 days, with AEROK off, and for 7 days with AEROK on in the absence of people.

### 2.6. Measurement of Temperature, Relative Humidity and CO_2_ Concentration

The microclimatic parameters were measured, during microbial monitoring, by Kimo AMI 310 STD device equipped with multifunction probes for temperature, relative humidity, and CO_2_. The measurements of temperature, relative humidity, and CO_2_ were performed within the same time frame as the microbial monitoring inside the room.

### 2.7. Instruments Calibration

All the samplers used were calibrated and certified by the manufacturers; in the case of the particle counter according to ISO 21501-4:2018 [63]. The pollen trap was regularly calibrated weekly with the flowmeter provided by the manufacturer according to ISO 16868:2019 [65].

### 2.8. Statistical Analysis

Statistical analysis was performed using the SPSS 26.0 package. Mean, standard deviation, minimum value, maximum value, and confidence intervals were calculated. For the comparison of bacterial and fungal contamination detected with AEROK off and with AEROK on, Student’s t-test was performed for paired data. In particular, the results of the 2nd hour were compared with the ones obtained in the 3rd hour and with the mean of the results of 3rd and 4th hour. Student’s *t*-test for independent data was used to assess the difference in particle concentration. Chi-square test was used to assess the difference between the concentration ratio I/O of pollen and fungal spore concentrations of *Alternaria* spp. with AEROK off and with AEROK on. *p* values < 0.05 were considered statistically significant.

## 3. Results

### 3.1. Microbial Air Sampling

#### 3.1.1. Bacteria

Table 1 shows the mean, standard deviation, and confidence interval of cfu/m^3^ for active sampling carried out by the DUOSAS sampler (mean values from the two DUOSAS samplers) over the 5 working days. The decrease in airborne bacterial contamination between the 2nd hour of sampling (AEROK off) and the 3rd hour (AEROK on) was 69%; between the 2nd hour and the mean between the values of the 3rd and 4th hour (AEROK on) was 70%.

Table 2 shows the cfu/m^3^ mean values of the MD8 sampler over the 5 working days. The decrease in airborne bacterial contamination between the 2nd hour of sampling (AEROK off) and the 3rd hour (AEROK on) was 56%; between the 2nd hour and the mean between the values of 3rd and 4th hour (AEROK on), it was 57%.

Regarding passive sampling, Figure 2 shows IMA values obtained at point E (center of the room). There was a decrease between the values of the 2nd hour (AEROK off) and the 3rd hour (AEROK on) of 44% and between the 2nd hour compared to the mean of 3rd and 4th hour (AEROK on) of 56%; moreover, IMA values obtained at the same time in the adjacent corridor in the absence of air treatment are shown.

Figure 3 shows bacterial air contamination values assessed by active sampling using DUOSAS and MD8, expressed as cfu/m^3^, and the results of passive sampling performed at point E, expressed as IMA.

Figure 4 shows the IMA median values and interquartile ranges referring to IMA values measured in the different sampling hours for each sampling point (Figure 1). At points A, B, and E a reduction in IMA values during the third and fourth hour (AEROK on) compared to the first two hours (AEROK off) was observed.

#### 3.1.2. Fungi

Fungal airborne contamination detected by DUOSAS significantly decreased between the 2nd hour (AEROK off) and the 3rd and 4th hour (AEROK on) (*p* = 0.013), from 27.1 cfu/m^3^ to 16.6 cfu/m^3^ (39% of reduction). Regarding the passive sampling carried out at point E, the fungal contamination decreased, although not significantly, between the 2nd hour (1.6 IMA) and the mean of the 3rd hour and the 4th hour (1.2 IMA).

#### 3.1.3. Door Openings

Table 3 shows the number of door openings per hour recorded during microbial sampling over 5 days of sampling, used as an indicator of the attendance in the room and adjacent spaces (*p* < 0.05). The highest number of openings was observed at the fourth hour.

### 3.2. Particles

Table 4 shows particles/m^3^ values according to the particle size assessed with AEROK off and with AEROK on during the 200 min of observation for each of the two conditions.

With AEROK turned on, a significant reduction of particles with diameters 1.0 and 2.0 μm was observed (*p* < 0.001). For particles with diameters ≥0.5, 5, 10, and 25 μm, no significant differences were found between the results obtained with AEROK off and with AEROK on.

### 3.3. Pollen and Fungal Spores

Figure 5 shows the mean values of the indoor-to-outdoor concentration ratio (I/O) of total pollen and fungal spores of *Alternaria* spp. Over the 7 days of sampling during the sampling period: 7 days with AEROK off and 7 days with AEROK on.

Significantly, the concentration ratio I/O of total pollen (*p* < 0.001) and spores of *Alternaria* spp. (*p* < 0.05) decreased with AEROK on.

### 3.4. Monitoring of Microclimatic Parameters

No significant changes in temperature, relative humidity, and CO_2_ values were observed between AEROK off and AEROK on.

## 4. Discussion

Microbial contamination of indoor environments is considered a public health concern due to the spread of pathogens or in general, aerosols harmful.

In recent years, portable (stand-alone) air purifiers have been increasingly adopted, also due to the SARS-CoV-2 pandemic, with widespread use in different confined spaces and not always sure of their effectiveness in reducing airborne contamination [31,66,67,68].

The characteristics of these devices can be very different in the mechanisms of action (HEPA filters, electrostatic decontamination, chemical, UV-C radiation, plasma, ozone, plasma, disinfection, etc., all the more so if it is a combination of different systems) [40,41,42,43,44,45,46,47,48,54]. Each mechanism of action has strengths and weaknesses (noise, high energy costs, pressure drop, disposal of chemicals, residues, or by-products). The goal has been to demonstrate the effectiveness of removing airborne contamination from particulate matter, viruses, bacteria, fungi, VOCs, allergens, etc.

Most of the studies have analyzed only one device at a time or a comparison between several devices with the same or different decontamination methods [32,33,34,35,36,37,38,39]. Often, the studies were limited to well-controlled laboratory situations or in rest conditions, but also field tests are important, because laboratory tests may not run with the complexities of the real situations (airflow patterns, occupant behaviors, interactions with the interior environment, etc.). A recent systematic review and meta-analysis [53] refers to 48 studies on stand-alone-air purifiers of which 36 performed in in residential buildings, 6 in health care settings, 4 in schools, and 2 in daycares. The research focused on pet allergens (11 papers), bioaerosols (9 papers), house dust mite allergen (8 papers), fungal spores (8 papers), SARS-CoV-2 RNA (5 papers), cockroach allergen (2 papers), pollen (2 papers), and airborne endotoxin (1 paper). The air-cleaning technologies applied in these studies were different, but a majority focused on HEPA filters. The tests have often been carried out without a vision of standardization regarding the methods used and the contaminants considered [21,32,33,34,35,36,37,38,39].

Based on previous experiences [69,70,71,72,73,74], we evaluated the effectiveness of an air treatment device with a global approach including the assessment of microbial, particle, pollen, and fungal contamination.

As for microbial contamination, active and passive sampling were used to have a complete assessment of air biological quality. They have different purposes: active sampling provides information about airborne viable particle concentration, whereas passive sampling measures the rate at which airborne viable particles settle on surfaces. Passive sampling provides a measure of the contribution of airborne contamination to the contamination of the surfaces.The results obtained highlight the effectiveness of the tested device in reducing airborne microbial contamination. This effectiveness is also supported by the observation of the trend of microbial contamination detected in the room where the device was operating compared to that of the contamination detected in the adjacent corridor, with similar activity. The presence of people and their activity are the main factors that increase airborne microbial contamination [75,76,77]. Between the 1st and 2nd hour, in both environments a reduction in microbial contamination was observed, corresponding to the reduction of administrative activity; during the 3rd and 4th hours, in correspondence with the increased administrative activity, an increase in microbial contamination was observed in the corridor, while in the room with AEROK on, microbial contamination continued to decrease. Regarding passive sampling, no reduction in airborne microbial contamination was observed at points C and D, contrary to what was observed at point E, probably affected, respectively, by the presence of a copier in continuous operation and the opening of the door.AEROK was also effective in reducing airborne particle contamination for particles with diameters ≥1.0 and 2.0 μm.Moreover, AEROK reduced the concentration of airborne pollen (*p* < 0.001) and fungal spores of *Alternaria* spp. (*p* < 0.05). This result is particularly relevant because it is obtained with windows constantly open and therefore in a condition that favors the replenishment of pollen and *Alternaria* spores.The tested device does not use chemical compounds with biocidal activity, resulting in the absence of chemical by-products to be disposed of. Therefore, the need for maintenance activities for filter cleaning is reduced, minimizing the risk of environmental contamination or exposure of personnel. The effectiveness of air-handling devices is usually assessed through laboratory tests and not through field studies within living or working environments, as carried out in this study. In addition, only some of the aspects considered in this study (bacterial, fungal, particulate contamination, pollen, and fungal spores) are often analyzed. Based on the literature, no converged conclusions can be drawn regarding the effectiveness of air purification technologies in practice. The documented and well-controlled laboratory studies do not adequately represent the practical situation in which the purifier systems are used [40]. Air cleaners are generally found to be effective in removing PM2.5 and PM10, with mean aggregated reductions of 49% and 44%, respectively [59]. These findings are in line with our results using an AEROK electrostatic copper filter device showing a slightly higher reduction of microbial contamination, and pollen and fungal spores I/O ratio.

## 5. Conclusions

This study provided an assessment of the effectiveness of the AEROK copper electro-static filtration system for air disinfection in a working environment under real working conditions. It can also represent a useful contribution to the definition of a standardized model for evaluating the effectiveness of devices to be used for air disinfection.

## Figures and Tables

**Figure 1 ijerph-21-01200-f001:**
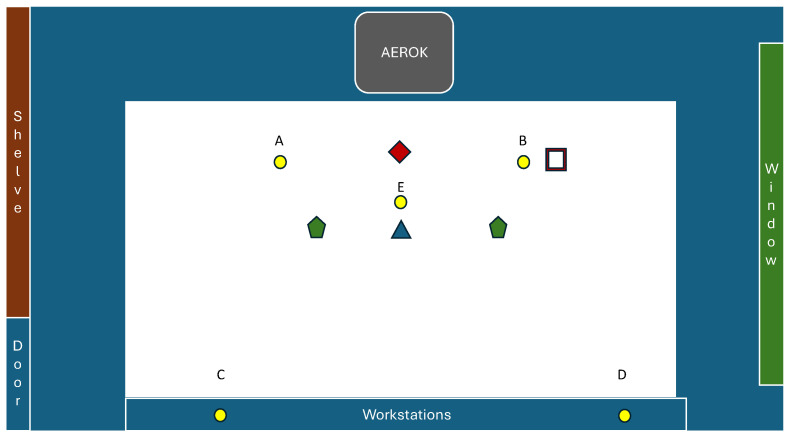
Positioning points of air samplers (active sampling with 
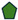
 DUOSAS, 
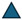
 MD8), 
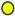
 Petri dishes (passive sampling), particle counter (
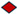
), pollen trap (
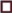
), and “AEROK” in the studied room. Particle sampler and pollen trap were used at different times than active and passive microbial sampling.

**Figure 2 ijerph-21-01200-f002:**
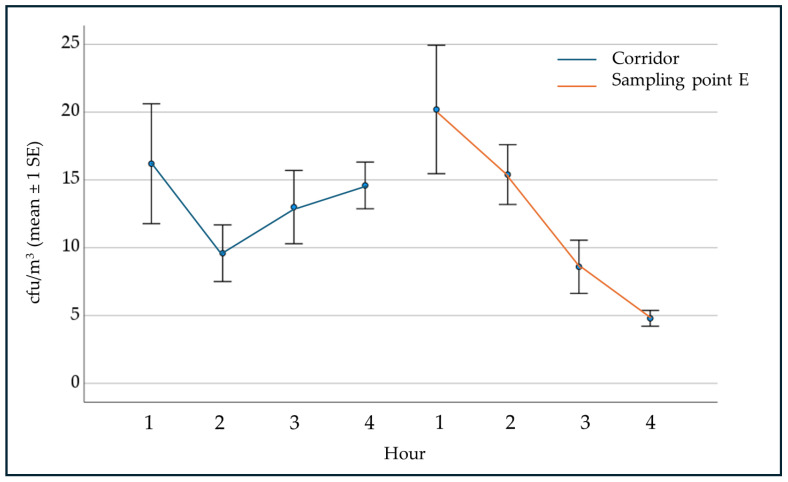
Bacterial air contamination (IMA) at point E and at corridor.

**Figure 3 ijerph-21-01200-f003:**
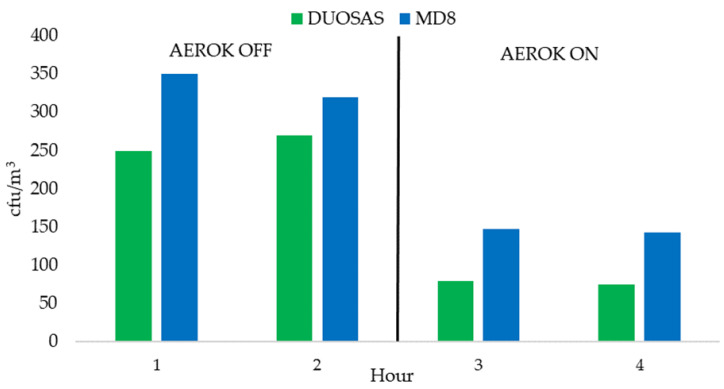
Cfu/m^3^ mean values obtained by DUOSAS and MD8.

**Figure 4 ijerph-21-01200-f004:**
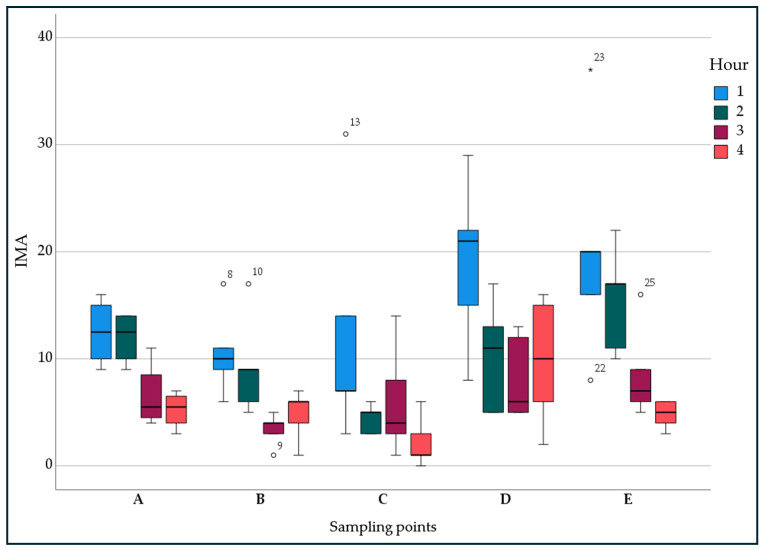
IMA median values and interquartile range at the different sampling points and times. ° Outliers; * Extreme outliers.

**Figure 5 ijerph-21-01200-f005:**
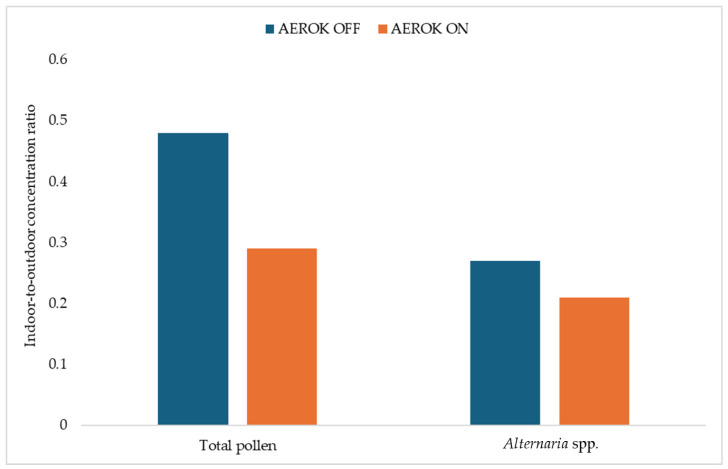
Mean values of the weekly indoor-to-outdoor concentration ratio of total pollen and fungal spores of *Alternaria* spp. with AEROK off and with AEROK on.

**Table 1 ijerph-21-01200-t001:** Bacterial air contamination (cfu/m^3^) assessed by DUOSAS samplers.

Hour	Mean	Standard Deviation	95% Confidence Interval	
			Upper	Lower
1	243.50	88.51	102.66	384.34
2	273.75	144.68	43.54	503.96
3	84.50	14.29	61,75	107.25
4	77.50	23.70	39.79	115.21

**Table 2 ijerph-21-01200-t002:** Bacterial air contamination (cfu/m^3^) assessed by MD8 sampler.

Hour	Mean	Standard Deviation	95% Confidence Interval	
			Upper	Lower
1	348.80	129.411	188.12	509.48
2	321.20	122.381	169.24	473.16
3	142.80	19.422	118.68	166.92
4	135.20	56.136	65.50	204.90

**Table 3 ijerph-21-01200-t003:** Number of door openings over 5 days of microbial sampling.

Hour	Mean	Standard Deviation	Minimum	Maximum
1	24.40	4.16	19	28
2	19.60	2.07	17	22
3	23.20	7.46	14	32
4	30.25	3.27	27	35

**Table 4 ijerph-21-01200-t004:** Maximum, minimum, mean, standard deviation of particles according to their size.

AEROK	Particle Diameter (μm) ≥	Numbers of Detections	Maximum	Minimum	Mean	St. Deviation
Off	0.5	200	3.81 × 10^6^	2.89 × 10^6^	3.29 × 10^6^	2.34 × 10^5^
	1		6.91 × 10^5^	1.96 × 10^5^	5.12 × 10^5^	1.34 × 10^5^
	2		2.66 × 10^5^	6.19 × 10^4^	1.39 × 10^5^	3.53 × 10^4^
	5		3.63 × 10^4^	5.50 × 10^2^	2.69 × 10^3^	5.62 × 10^3^
	10		1.94 × 10^4^	0	4.96 × 10^2^	1.89 × 10^3^
	25		5.70 × 10^2^	0	0.18 × 10^2^	0.70 × 10^2^
On	0.5	200	5.43 × 10^6^	4.32 × 10^5^	2.08 × 10^6^	8.60 × 10^5^
	1		1.57 × 10^6^	1.09 × 10^5^	4.81 × 10^5^	2.43 × 10^5^
	2		5.96 × 10^5^	3.16 × 10^4^	1.30 × 10^5^	8.79 × 10^4^
	5		3.92 × 10^4^	1.70 × 10^3^	1.86 × 10^3^	5.28 × 10^3^
	10		1.09 × 10^4^	0	3.43 × 10^2^	1.37 × 10^3^
	25		7.50 × 10^2^	0	0.16 × 10^2^	0.78 × 10^2^

## Data Availability

Data are available upon request.

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
