# Peer review of "Study on the Effectiveness of a Copper Electrostatic Filtration System “Aerok 1.0” for Air Disinfection"

_ijerph, 2024, doi:10.3390/ijerph21091200_

Round 1

Reviewer 1 Report

Comments and Suggestions for Authors

This paper presents a novel pollutant treatment device and conducts a study on its filtration characteristics towards relevant contaminants, which holds significant engineering implications and value. However, there are several aspects that require improvement to address the existing shortcomings.

  1. While the authors mention the potential value of portable devices as preventive tools during the SARS-CoV-2 pandemic, across various environments, there seems to be a lack of research specifically targeting the SARS-CoV-2 virus.
  2. The introduction section inadequately reviews the work conducted by other scholars in this field, leaving gaps in the literature context.
  3. The study focuses solely on exploring a single device without comparing its performance against existing equipment, which could provide valuable insights into its advantages and limitations.
  4. It is unclear whether the relevant testing equipment has undergone calibration, which is crucial for ensuring the accuracy and reliability of the experimental results.
  5. The presentation of graphs and charts could be optimized for better aesthetics and clarity, enhancing the readability and comprehensiveness of the data.
  6. The conclusions should be structured in a bullet-point format, outlining the quantitative research findings of this study in a concise and organized manner. This will facilitate the understanding and application of the research outcomes.
Comments on the Quality of English Language

Minor editing of English language required.

Author Response

We would like to thank the Referees for their precious suggestions, which will certainly contribute to the improvement of the manuscript, and we respond point by point to their requests or suggestions.

Referee 1

This paper presents a novel pollutant treatment device and conducts a study on its filtration characteristics towards relevant contaminants, which has significant engineering implications and value. However, there are several aspects that require improvement to address existing shortcomings.

While the authors mention the potential value of portable devices as preventive tools during the SARS-CoV-2 pandemic, in various settings, there seems to be a lack of research specifically targeting the SARS-CoV-2 virus.

The reference to the SARS-CoV-2 pandemic relates to the need for air disinfection in healthcare and non-healthcare environments, leading to an impetus towards the provision of suitable instrumentation that, due to its characteristics, can be a useful tool for air disinfection not only against viruses, but also against other airborne microorganisms and, in general, bioaerosols.  We have chosen to focus on airborne bacteria, fungi, particles, pollen and spore of Alternaria and not on viruses’ contamination.

The introductory section inadequately reviews the work conducted by other scholars in this field, leaving gaps in the context of the literature.

The introduction has been implemented including some references dealing with the different air treatment systems used.

The study focuses solely on the exploration of a single device without comparing its performance with existing equipment, which could provide valuable insights into its advantages and limitations.

Our aim was not to compare AEROK with other devices, but to evaluate its effectiveness in reducing airborne contamination and, at the same time, to propose a standardized multi-sectoral approach to this type of study.

It is unclear whether the relevant test equipment has undergone calibration, which is critical to ensure the accuracy and reliability of the experimental results. The presentation of graphs and charts could be optimized for better aesthetics and clarity, improving the readability and completeness of the data.

All of the samplers/devices used were calibrated and certified by the manufacturers; in the case of the particle counter according to ISO 21501-4:2018. The pollen trap was regularly calibrated weekly with the flowmeter provided by manufacturer according to ISO 16868:2019.

A paragraph on the calibration of the instrumentation used in the study was added.

We have modified graphs and tables to make them easier to read.

The conclusions should be structured in a bulleted list format, outlining the quantitative research findings of this study in a concise and organized manner. This will facilitate understanding and application of the research findings.

We have used the bulleted list as suggested.

Sincerely

Reviewer 2 Report

Comments and Suggestions for Authors

 The study of Albertini et al. reports about the effectiveness of the air purifying device on airborne bioaerosol or total particle count. The study is a useful contribution to the area as it addresses several methodological aspects often overlooked in similar studies. However, the paper should be improved on some key data analysis aspects as well as English of the paper before it can be accepted for publication.

More significant comments

Introduction should be improved, especially in terms of what this study is aiming considering other similar papers.

Abstract and throughout the paper: Significant digits denote precision. I doubt that the precision was better than 1% (typically not better than 5% in aerosol measurements, hence, only keeping 2 significant digits (like 55% and 69%) is justified. Same applies to standard deviation and other values. Table 6 implies that every single particle was counted while that was not the case given standard deviation. Change to scientific notation, e.g. 3.8x10^6.

AEROK is the mysterious device as I could not search for it and should, therefore, be described in more detail or valid reference given.

All Figures lack the error bars/sticks. The Tables include them, but not Figures. Also, there is no justification keeping both, Tables and Figures, carrying the same information.

Tables formatting is untidy: lines, comma/dots, bold/regular font.

Figures have missing legends to indicate separate data lines. Room drawings have ample space for noting devices or specific spots.

Other comments

Section 2.5.2. It is unclear when windows were opened closed with respect to every studied species. What was the point of keeping windows opened and AEROK ON? What effect is tested here when indoor particles are constantly replenished and partially removed by AEROK?

It is unclear how the data were treated. Were the same hours averaged over 5 days? Make that clear.

Discussion paragraph 1. Unclear sentence. What does the "wide diffusion in different environments mean? Second sentence is also unclear.

Example papers should be properly referenced before counting, counting alone is not enough.

Line 277. Accumulation or replenishment?

Comments on the Quality of English Language

There are several unclear sentences, typos as noted above.

Author Response

We would like to thank the Reviewers for their precious suggestions, which will certainly contribute to the improvement of the manuscript, and we respond point by point to their requests or suggestions.

Reviewer 2

The study by Albertini et al. reports the effectiveness of the air purification device on atmospheric bioaerosol or total particle count. The study is a useful contribution to the area as it addresses several methodological aspects that are often overlooked in similar studies. However, the paper should be improved on some key aspects of data analysis and the English of the paper before it can be accepted for publication.

Most significant comments

The introduction should be improved, especially in terms of what this study aims to achieve considering other similar papers.

The introduction has been modified to better clarify the purpose of the study.

Abstract and throughout the paper: significant figures indicate precision. I doubt that the accuracy was better than 1% (typically not better than 5% in aerosol measurements, so there is justification for keeping only 2 significant figures (such as 55% and 69%).

We have followed the suggestion

The same goes for the standard deviation and other values. Table 6 implies that every single particle was counted, whereas this was not the case given the standard deviation. Switch to scientific notation, such as 3.8x10^6.

We have followed the suggestion

AEROK is the mystery device because I could not look it up and should therefore be described in more detail or a valid reference should be provided.

We have added the following wording in the text: 'AEROK 1.0®' (AERSAFE s.r.l Trento, Italy).

All Figures are free of error bars/sticks. Tables include them, but not Figures. Also, there is no justification for keeping both Tables and Figures containing the same information.

We have deleted Figure 3 and Tables 3 and 4. In the Figures we added error bars/sticks if missing.

Tables formatting is messy: lines, commas/periods, bold/regular font.

We have formatted Table 6, now Table 4.

Figures have missing legends to indicate separate lines of data. Room drawings have ample space to note specific devices or points.

We have eliminated Figure 2 by adding the information that was in Figure 1.

Other comments

Section 2.5.2. It is unclear when windows were opened closed with respect to every studied species.

We have modified the text to clarify this point.

What was the point of keeping windows opened and AEROK ON? What effect is tested here when indoor particles are constantly replenished and partially removed by AEROK?

The open windows were intended to allow the replenishment of fungal spores and pollen from outside, considering that all airborne pollen and Alternaria spores (in the absence of an indoor source) are from outdoor origin.

It is unclear how the data were treated. Were the same hours averaged over 5 days? Make that clear.

Alternaria spores and pollen were sampled continuously for 7 days, 24 hours a day, with AEROK switched on and in the same way with AEROK switched off.

Discussion paragraph 1. Unclear sentence. What does the "wide diffusion in different environments mean?

We have modified the sentence to make it more understandable.

Second sentence is also unclear.

We have modified the sentence to make it more understandable.

Example papers should be properly referenced before counting, counting alone is not enough.

We have cited the references that we considered important in the context of the topic and referenced some of them more directly related to what was discussed.

Line 277. Accumulation or replenishment?

We have corrected the sentence using the word “replenishment”.

The english was revised by a native speaker; the manuscript was checked for typing errors.

Sincerely